# Breaking certified defenses: Semantic adversarial examples with spoofed robustness certificates

**Amin Ghiasi** [*], **Ali Shafahi**[*] **& Tom Goldstein**
University of Maryland
{amin,ashafahi,tomg}@cs.umd.edu

## Abstract

To deflect adversarial attacks, a range of "certified" classifiers have been proposed. In addition to labeling an image, certified classifiers produce (when possible) a certificate guaranteeing that the input image is not an $\ell_p$-bounded adversarial example. We present a new attack that exploits not only the labelling function of a classifier, but also the certificate generator. The proposed method applies large perturbations that place images far from a class boundary while maintaining the imperceptibility property of adversarial examples. The proposed "Shadow Attack" causes certifiably robust networks to mislabel an image and simultaneously produce a "spoofed" certificate of robustness.

## 1 Introduction

Conventional training of neural networks has been shown to produce classifiers that are highly sensitive to adversarial perturbations (Szegedy et al., 2013; Biggio et al., 2013), "natural looking" images that have been manipulated to cause misclassification by a neural network (Figure 1). While a wide range of defenses exist that harden neural networks against such attacks (Madry et al., 2017; Shafahi et al., 2019), defenses based on heuristics and tricks are often easily breakable Athalye et al. (2018). This has motivated work on *certifiably* secure networks — classifiers that produce a label for an image, and also (when possible) a rigorous guarantee that the input is not adversarially manipulated (Cohen et al., 2019; Zhang et al., 2019b).

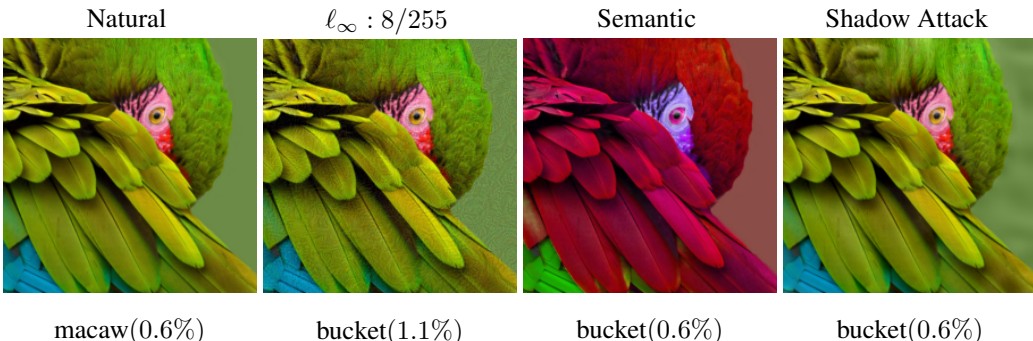

| Natural | $\ell_\infty : 8/255$ | Semantic | Shadow Attack |
|---|---|---|---|
| macaw(0.6%) | bucket(1.1%) | bucket(0.6%) | bucket(0.6%) |

Figure 1: All adversarial examples have the goal of fooling classifiers while looking "natural". Standard attacks limit the $\ell_p$-norm of the perturbation, while semantic attacks have large $\ell_p$-norm while still producing natural looking images. Our attack produces large but visually subtle semantic perturbations that not only cause misclassification, but also cause a certified defense to issue a "spoofed" high-confidence certificate. In this case, a certified Gaussian smoothing classifier mislabels the image, and yet issues a certificate with radius 0.24, which is a larger certified radius than its corresponding unmodified ImageNet image which is 0.13.

---

[*]equal contribution

To date, all work on certifiable defenses has focused on deflecting $\ell_p$-bounded attacks, where $p = 2$ or $\infty$ (Cohen et al., 2019; Gowal et al., 2018; Wong et al., 2018). After labelling an image, these defenses then check whether there exists an image of a different label within $\epsilon$ units (in the $\ell_p$ metric) of the input, where $\epsilon$ is a security parameter chosen by the user. If the classifier assigns all images within the $\epsilon$ ball the same class label, then a certificate is issued, and the input image known not to be an $\ell_p$ adversarial example.

In this work, we demonstrate how a system that relies on certificates as a measure of label security can be exploited. We present a new class of adversarial examples that target not only the classifier output label, *but also the certificate.* We do this by adding adversarial perturbations to images that are large in the $\ell_p$ norm (larger than the $\epsilon$ used by the certificate generator), and produce attack images that are surrounded by a large ball exclusively containing images of the same label. The resulting attacks produce a "spoofed" certificate with a seemingly strong security guarantee despite being adversarially manipulated. Note that the statement made by the certificate (i.e., that the input image is not an $\epsilon$ adversarial example in the chosen norm) is still technically correct, however in this case the adversary is hiding behind a certificate to avoid detection by a certifiable defense.

In summary, we consider methods that attack a certified classifier in the following sense:

- **Imperceptibility:** the adversarial example "looks like" its corresponding natural base example,
- **Misclassification:** the certified classifier assigns an incorrect label to the adversarial example, and
- **Strongly certified:** the certified classifier provides a strong/large-radius certificate for the adversarial example.

While the existence of such an attack does not invalidate the certificates produced by certifiable systems, it should serve as a warning that certifiable defenses are not inherently secure, and one should take caution when relying on certificates as an indicator of label correctness.

## BACKGROUND

In the white-box setting, where the attacker knows the victim's network and parameters, adversarial perturbation are often constructed using first-order gradient information (Carlini & Wagner, 2017; Kurakin et al., 2016; Moosavi-Dezfooli et al., 2016) or using approximations of the gradient (Uesato et al., 2018; Athalye et al., 2018). The prevailing formulation for crafting attacks uses an additive adversarial perturbation, and perceptibility is minimized using an $\ell_p$-norm constraint. For example, $\ell_\infty$-bounded attacks limit how much each pixel can move, while $\ell_0$ adversarial attacks limit the number of pixels that can be modified, without limiting the size of the change to each pixel (Wiyatno & Xu, 2018).

It is possible to craft imperceptible attacks without using $\ell_p$ bounds (Brown et al., 2018). For example, Hosseini & Poovendran (2018) use shifting color channels, Wong et al. (2019) use the Wasserstein ball/distance, and Engstrom et al. (2017) use rotation and translation to craft "semantic" adversarial examples. In Figure 1, we produce semantic adversarial examples using the method of Hosseini & Poovendran (2018) which is a greedy approach that transforms the image into HSV space, and then, while keeping V constant, tries to find the smallest S perturbation causing misclassification[1]. Other variants use generative models to construct natural looking images causing misclassification (Song et al., 2018; Dunn et al., 2019).

In practice, many of the defenses which top adversarial defense leader-board challenges are non-certified defenses (Madry et al., 2017; Zhang et al., 2019a; Shafahi et al., 2019). The majority of these defenses make use of *adversarial training*, in which attack images are crafted during training and injected into the training set. These non-certified defenses are mostly evaluated against PGD-based attacks, resulting in an upper-bound on robustness.

Certified defenses, on the other-hand, provably make networks resist $\ell_p$-bounded perturbations of a certain radius. For instance, randomized smoothing (Cohen et al., 2019) is a certifiable defense against $\ell_2$-norm bounded attacks, and CROWN-IBP (Zhang et al., 2019b) is a certifiable defense

---

[1]In fig. 1, the adversarial example has saturation=0

against $l_\infty$-norm bounded perturbations. Both of these defenses produce a class label, and also a guarantee that the image could not have been crafted by making small perturbations to an image of a different label. Certified defenses can also benefit from adversarial training. For example, Salman et al. (2019) recently improved the certified radii of randomized smoothing (Cohen et al., 2019) by training on adversarial examples generated for the smoothed classifier.

To the best of our knowledge, prior works have focused on making adversarial examples that satisfy the imperceptibility and misclassification conditions, but none have investigated manipulating certificates, which is our focus here.

The reminder of this paper is organized as follows. In section 2 we introduce our new approach Shadow Attack for generating adversarial perturbations. This is a hybrid model that allows various kinds of attacks to be compounded together, resulting in perturbations of large radii. In section 3 we present an attack on "randomized smoothing" certificates (Cohen et al., 2019). Section 4 shows an ablation study which illustrates why the elements of the Shadow Attack are important for successfully manipulating certified models. In section 5 we generate adversarial examples for "CROWN-IBP" (Zhang et al., 2019b). Finally, we discuss results and wrap up in section 6.

## 2 THE SHADOW ATTACK

Because certificate spoofing requires large perturbations (larger than the $\ell_p$ ball of the certificate), we propose a simple attack that ensembles numerous modes to create large perturbations. Our attack can be seen as the generalization of the well-known PGD attack, which creates adversarial images by modifying a clean base image. Given a loss function $L$ and an $\ell_p$-norm bound $\epsilon$ for some $p \geq 0$, PGD attacks solve the following optimization problem:

$$\max_\delta L(\theta, x + \delta) \tag{1}$$

$$s.t. \ \|\delta\|_p \leq \epsilon, \tag{2}$$

where $\theta$ are the network parameters and $\delta$ is the adversarial perturbation to be added to the clean input image $x$. Constraint 2 promotes imperceptibility of the resulting perturbation to the human eye by limiting the perturbation size. In the shadow attack, instead of solving the above constrained optimization problem, we solve the following problem with a range of penalties:

$$\max_\delta L(\theta, x + \delta) - \lambda_c C(\delta) - \lambda_{tv} TV(\delta) - \lambda_s Dissim(\delta), \tag{3}$$

where $\lambda_c, \lambda_{tv}, \lambda_s$ are scalar penalty weights. Penalty $TV(\delta)$ forces the perturbation $\delta$ to have small total variation ($TV$), and so appear more smooth and natural. Penalty $C(\delta)$ limits the perturbation $\delta$ globally by constraining the change in the mean of each color channel $c$. This constraint is needed since total variation is invariant to constant/scalar additions to each color channel, and it is desirable to suppress extreme changes in the color balances of images.

Penalty $Dissim(\delta)$ promotes perturbations $\delta$ that assume similar values in each color channel. In the case of an RGB image of shape $3 \times W \times H$, if $Dissim(\delta)$ is small, the perturbations to red, green, and blue channels are similar, i.e., $\delta_{R,w,h} \approx \delta_{G,w,h} \approx \delta_{B,w,h}, \forall (w, h) \in W \times H$. This amounts to making the pixels darker/lighter, without changing the color balance of the image. Later, in section 3, we suggest two ways of enforcing such similarity between RGB channels and we find both of them effective:

- **1-channel attack** strictly enforces $\delta_{R,i} \approx \delta_{G,i} \approx \delta_{B,i}, \forall i$ by using just one array to simultaneously represent each color channel $\delta_{W \times H}$. On the forward pass, we duplicate $\delta$ to make a 3-channel image. In this case, $Dissim(\delta) = 0$, and the perturbation is greyscale.

- **3-channel attack** uses a 3-channel perturbation $\delta_{3 \times W \times H}$, along with the dissimilarity metric $Dissim(\delta) = \|\delta_R - \delta_B\|_p + \|\delta_R - \delta_G\|_p + \|\delta_B - \delta_G\|_p$.

All together, the three penalties minimize perceptibility of perturbations by forcing them to be $(a)$ small, $(b)$ smooth, and $(c)$ without dramatic color changes (e.g. swapping blue to red). At the same time, these penalties allow perturbations that are very large in $\ell_p$-norm.

## 2.1 Creating untargeted attacks

We focus on spoofing certificates for *untargeted* attacks, in which the attacker does not specify the class into which the attack image moves. To achieve this, we generate an adversarial perturbation for all possible wrong classes $\bar{y}$ and choose the best one as our strong attack:

$$\max_{\bar{y} \neq y, \delta} -L(\theta, x + \delta \| \bar{y}) - \lambda_c C(\delta) - \lambda_{tv} TV(\delta) - \lambda_s Dissim(\delta) \qquad (4)$$

where $y$ is the true label/class for the clean image $x$, and $L$ is a spoofing loss that promotes a strong certificate. We examine different choices for $L$ for different certificates below.

## 3 Attacks on Randomized Smoothing

The Randomized Smoothing method, first proposed by Lecuyer et al. (2018) and later improved by Li et al. (2018), is an adversarial defense against $\ell_2$-norm bounded attacks. Cohen et al. (2019) prove a tight robustness guarantee under the $\ell_2$ norm for smoothing with Gaussian noise. Their study was the first certifiable defense for the ImageNet dataset (Deng et al., 2009). The method constructs certificates by first creating many copies of an input image contaminated with random Gaussian noise of standard deviation $\sigma$. Then, it uses a base classifier (a neural net) to make a prediction for all of the images in the randomly augmented batch. Depending on the level of the consensus of the class labels at these random images, a certified radius is calculated that can be at most $4\sigma$ (in the case of perfect consensus).

Intuitively, if the image is far away from the decision boundary, the base classifier should predict the same label for each noisy copy of the test image, in which case the certificate is strong. On the other hand, if the image is adjacent to the decision boundary, the base classifier's predictions for the Gaussian augmented copies may vary. If the variation is large, the smoothed classifier abstains from making a prediction.

To spoof strong certificates (large certified radius) for an incorrect class, we must make sure that the majority of a batch of noisy images around the adversarial image are assigned the same (wrong) label. We do this by minimizing the cross entropy loss relative to a chosen (incorrect) label, averaged over a large set of randomly perturbed images. To this end, we minimize equation 4, where $L$ is chosen to be the average cross-entropy over a batch of Gaussian perturbed copies. This method is analogous to the technique presented by Shafahi et al. (2018) for generating universal perturbations that are effective when added to a large number of different images.

### Results

Cohen et al. (2019) show the performance of the Gaussian smoothed classifier on CIFAR-10 (Krizhevsky et al.) and ImageNet (Deng et al., 2009). To attack the CIFAR-10 and ImageNet smoothed classifiers, we use 400 randomly sampled Gaussian images, $\lambda_{tv} = 0.3$, $\lambda_c = 1.0$, and perform 300 steps of SGD with learning rate 0.1. We choose the functional regularizers $C(\delta)$ and $TV(\delta)$ to be

$$C(\delta) = \| \text{Avg}(|\delta_R|), \text{Avg}(|\delta_G|), \text{Avg}(|\delta_B|) \|_2^2, \quad \text{and} \quad TV(\delta_{i,j}) = \text{anisotropic-TV}(\delta_{i,j})^2,$$

where $| \cdot |$ is the element-wise absolute value operator, and $\text{Avg}$ computes the average. For the $Dissim$ regularizer, we experiment with both the 1-Channel attack that ensures $Dissim(\delta) = 0$, and the 3-Channel attack by setting $Dissim(\delta) = \|(\delta_R - \delta_G)^2, (\delta_R - \delta_B)^2, (\delta_G - \delta_B)^2\|_2$ and $\lambda_s = 0.5$. For the validation examples on which the smoothed classifier does not abstain (see Cohen et al. (2019) for more details), the less-constrained 3-channel attack is always able to find an adversarial example while the 1-channel attack also performs well, achieving 98.5% success. [2] In section 4 we will discuss in more detail other differences between 1-channel and 3-channel attacks. The results are summarized in Table 1. For the various base-models and choices of $\sigma$, our adversarial examples are able to produce certified radii that are on average larger than the certified radii produced for natural images. For ImageNet, since attacking all 999 remaining target classes is computationally expensive, we only attacked target class IDs 100, 200, 300, 400, 500, 600, 700, 800, 900, and 1000.

---

[2]Source code for all experiments can be found at: https://github.com/AminJun/BreakingCertifiableDefenses

Table 1: Certified radii produced by the Randomized Smoothing method for Shadow Attack images and also natural images (larger radii means a stronger/more confident certificate).

| Dataset | $\sigma(l_2)$ | Unmodified/Natural Images | | Shadow Attack | |
|---|---|---|---|---|---|
| | | Mean | STD | Mean | STD |
| CIFAR-10 | 0.12 | 0.14 | 0.056 | **0.22** | 0.005 |
| | 0.25 | 0.30 | 0.111 | **0.35** | 0.062 |
| | 0.50 | 0.47 | 0.234 | **0.65** | 0.14 |
| | 1.00 | 0.78 | 0.556 | **0.85** | 0.442 |
| ImageNet | 0.25 | 0.30 | 0.109 | **0.31** | 0.109 |
| | 0.50 | 0.61 | 0.217 | 0.38 | 0.191 |
| | 1.00 | 1.04 | 0.519 | 0.64 | 0.322 |

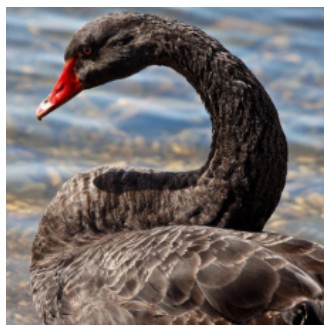 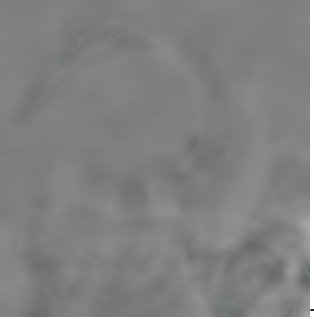 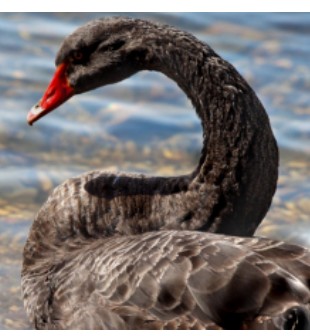

(a) Natural image $(x)$       (b) Adversarial perturbation $(\delta)$       (c) Adversarial example $(x + \delta)$

Figure 2: An adversarial example built using our Shadow Attack for the smoothed ImageNet classifier for which the certifiable classifier produces a large certified radii. The adversarial perturbation is smooth and natural looking even-though it is large when measured using $\ell_p$-metrics. Also see Figure 16 in the appendix.

Figure 2 depicts a sample adversarial example built for the smoothed ImageNet classifier that produces a strong certificate. The adversarial perturbation causes the batch of Gaussian augmented black swan images to get misclassified as hooks. For more, see appendix 16.

## 4   ABLATION STUDY OF THE ATTACK PARAMETERS

In this section we perform an ablation study on the parameters of the Shadow Attack to evaluate $(i)$ the number of SGD steps needed, $(ii)$ the importance of $\lambda_s$ (or alternatively using 1-channel attacks), and $(iii)$ the effect of $\lambda_{tv}$.

The default parameters for all of the experiments are as follows unless explicitly mentioned: We use 30 SGD steps with learning rate 0.1 for the optimization. All experiments except part $(ii)$ use 1-channel attacks for the sake of simplicity and efficiency (since it has less parameters). We assume $\lambda_{tv} = 0.3$, $\lambda_c = 20$, and use batch-size 50. We present results using the first example from each class of the CIFAR-10 validation set.

Figure 3 shows how the adversarial example evolves during the first few steps of optimization (See appendix 13 for more examples). Also, figures 4, 5, and 6 show the evolution of $L(\delta)$, $TV(\delta)$, and $C(\delta)$, respectively (Note that we use 1-channel attacks, so $Dissim(\delta)$ is always 0). We find that taking just 10 SGD steps is enough for convergence on CIFAR-10, but for our main results (i.e. attacking Randomized Smoothing in section 3 and attacking CROWN-IBP in section 5) we take 300 steps to be safe.

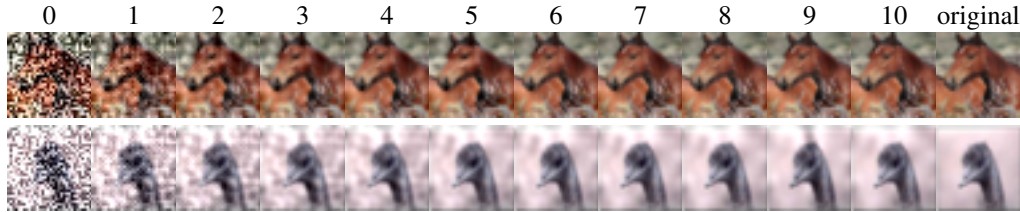

Figure 3: The first 10 steps of optimization (beginning with a randomly perturbed image copy).

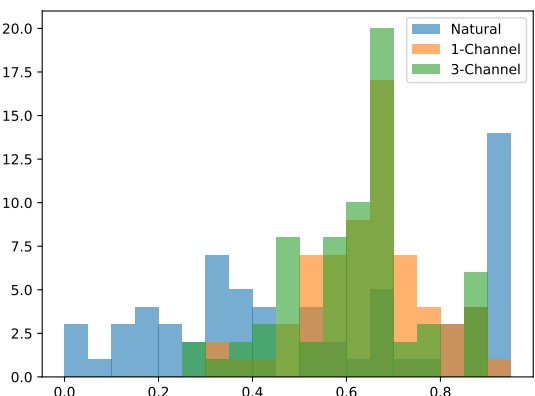

Figure 9: Histogram of randomized smoothed certificate radii for 100 randomly sampled CIFAR-10 validation images vs those calculated for their adversarial examples crafted using our 1-channel and 3-channel adversarial Shadow Attack attacks. The "robust" victim classifier is based off Resnet-110, and smoothed with $\sigma = 0.50$. 1-channel attacks are almost as good as the less-restricted 3-channel attacks.

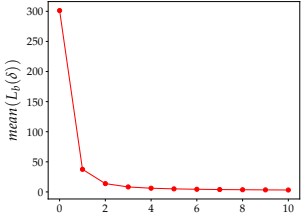

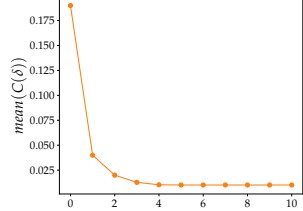

Figure 4: Average $L_b(\delta)$ in the first 10 steps.

Figure 5: Average $TV(\delta)$ in the first 10 steps.

Figure 6: Average $C(\delta)$ in the first 10 steps.

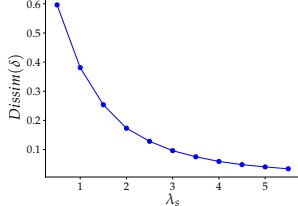

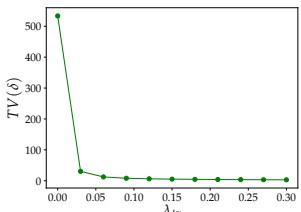

Figure 7: The effect of $\lambda_s$ on the resulting $Dissim(\delta)$

Figure 8: The effect of $\lambda_{tv}$ on the resulting $TV(\delta)$

To explore the importance of $\lambda_s$, we use 3-channel attacks and vary $\lambda_s$ to produce different images in figure 11[3].

---

[3]See figure 14 in the appendix for more examples.

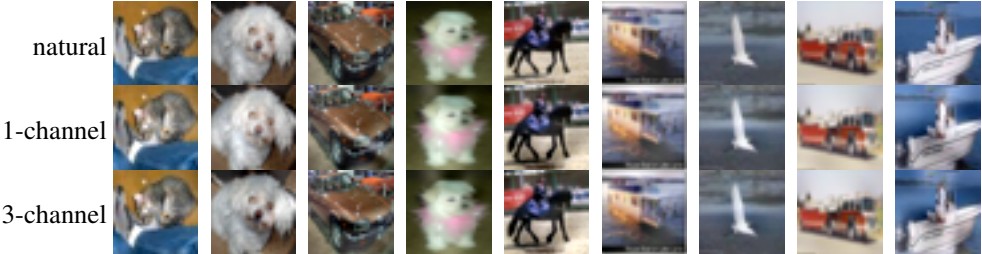

Figure 10: The visual effect of Shadow Attack on 9 randomly selected CIFAR-10 examples using 1-Channel and 3-Channel attacks.

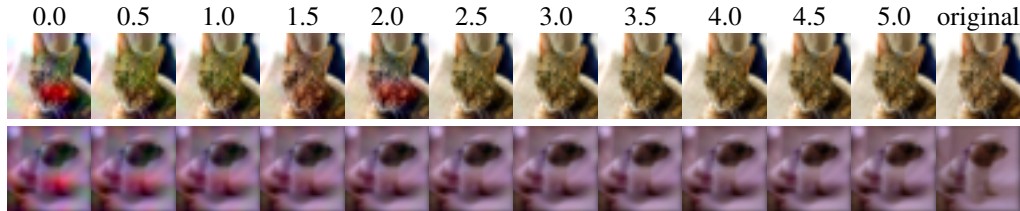

Figure 11: The visual effect of $\lambda_s$ on perceptibility of the perturbations. The first row shows the value of $\lambda_s$.

Also, figure 7 shows the mean $Dissim(\delta)$ for different values of $\lambda_s (0 \leq \lambda_s \leq 5.0)$. We also plot the histogram of the certificate radii in figure 9. Figure 10 compares 1-Channel vs 3-Channel attacks for some of randomly selected CIFAR-10 images. Finally, we explore the effect of $\lambda_{tv}$ on imperceptibility of the perturbations in Figure 12. See table 15 for more images, and figure 8 for the impact of parameters on $TV(\delta)$.

# 5 ATTACKS ON CROWN-IBP

Interval Bound Propagation (IBP) methods have been recently studied as a defense against $\ell_\infty$-bounded attacks. Many recent studies such as Gowal et al. (2018); Xiao et al. (2018); Wong et al. (2018); Mirman et al. (2018) have investigated IBP methods to train provably robust networks. To the best of our knowledge, the CROWN-IBP method by Zhang et al. (2019b) achieves state-of-the-art performance for MNIST (LeCun & Cortes, 2010), Fashion-MNIST (Xiao et al., 2017), and CIFAR-10 datasets among certifiable $\ell_\infty$ defenses. In this section we focus on attacking Zhang et al. (2019b) using CIFAR-10.

IBP methods (over)estimate how much a small $\ell_\infty$-bounded noise in the input can impact the classification layer. This is done by propagating errors from layer to layer, computing bounds on the maximum possible perturbation to each activation. During testing, the user chooses an $\ell_\infty$ perturbation bound $\epsilon$, and error propagation is used to bound the magnitude of the largest achievable perturbation in network output. If the output perturbation is not large enough to flip the image label, then a certificate is produced. If the output perturbation is large enough to flip the image label,

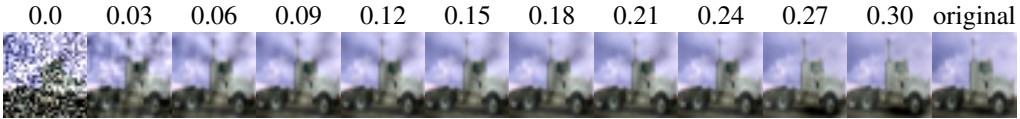

Figure 12: The visual effect of $\lambda_{tv}$ on the on imperceptibility of the perturbations. The first row shows the value of $\lambda_{tv}$

Table 2: "Robust error" for natural images, and "attack error" for Shadow Attack images using the CIFAR-10 dataset, and CROWN-IBP models. Smaller is better.

| $\epsilon(l_\infty)$ | Model Family | Method | Robustness Errors | | |
| --- | --- | --- | --- | --- | --- |
| | | | Min | Mean | Max |
| 2/255 | 9 small models | CROWN-IPB | 52.46 | 57.55 | 60.67 |
| | | Shadow Attack | **45.90** | **53.89** | 65.74 |
| | 8 large models | CROWN-IBP | 52.52 | 53.9 | 56.05 |
| | | Shadow Attack | **46.21** | **49.77** | **51.79** |
| 8/255 | 9 small models | CROWN-IBP | 71.28 | 72.15 | 73.66 |
| | | Shadow Attack | **63.43** | **66.94** | **71.02** |
| | 8 large models | CROWN-IBP | 70.79 | 71.17 | 72.29 |
| | | Shadow Attack | **64.04** | **67.32** | **71.16** |

then a certificate is not produced. During network training, IBP methods include a term in the loss function that promotes tight error bounds that result in certificates. Our attack directly uses the loss function term used during IBP network training in addition to the Shadow Attack penalties; we search for an image that is visually similar to the base image, while producing a bound on the output perturbation that is too small to flip the image label. Note that there is a subtle difference between crafting conventional adversarial examples which ultimately targets misclassification and our adversarial examples which aim to produce adversarial examples which cause misclassification *and* produce strong certificates. In the former case, we only need to attack the cross-entropy loss. If we use our Shadow Attack to craft adversarial examples based on the cross-entropy loss, the robustness errors are on average roughly 50% larger than those reported in table 2 (i.e., simple adversarial examples produce weaker certificates.)

We attack 4 classes of networks released by Zhang et al. (2019b) for CIFAR-10. There are two classes of IBP architectures, one of them consists of 9 small models and the other consists of 8 larger models. For each class of architecture, there are two sets of pre-trained models: one for $\epsilon = 2/255$ and one for $\epsilon = 8/255$. We use $\lambda_{tv} = 0.000009$, $\lambda_c = 0.02$, $C(\delta) = \|\delta\|_2$ and set the learning rate to 200 and for the rest of the regularizers and hyper-paramters we use the same hyper-parameters and regularizers as in 3. For the sake of efficiency, we only do 1-channel attacks. We attack the 4 classes of models and for each class, we report the min, mean, and max of the robustness errors and compare them with those of the CROWN-IBP paper.

To quantify the success of our attack, we define two metrics of error. For natural images, we report the rate of "robust errors," which are images that are either (i) incorrectly labeled, or (ii) correctly labelled but without a certificate. In contrast, for attack images, we report the rate of "attack errors," which are either (i) correctly classified or (ii) incorrectly classified but without a certificate. Table 2 shows the robust error on natural images, and the attack error on Shadow Attack images for the CROWN-IBP models. With $\epsilon = 2/255$, our attack finds adversarial examples that certify roughly 15% of the time (i.e., attack error <85%). With $\epsilon = 8/255$, our attack finds adversarial examples that are incorrectly classified, and yet certify even more often than natural images.

## 6 CONCLUSION

We demonstrate that it is possible to produce adversarial examples with "spoofed" certified robustness by using large-norm perturbations. Our adversarial examples are built using our Shadow Attack that produces smooth and natural looking perturbations that are often less perceptible than those of the commonly used $\ell_p$-bounded perturbations, while being large enough in norm to escape the certification regions of state of the art principled defenses. This work suggests that the certificates produced by certifiably robust classifiers, while mathematically rigorous, are not always good indicators of robustness or accuracy.

**Acknowledgements:** Goldstein and his students were supported by the DARPA QED for RML program, the DARPA GARD program, and the National Science Foundation.

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

## A    APPENDIX

In this section, we include the complete results of our ablation study. As we mentioned in section 4, we use is a subset of CIFAR-10 dataset, including one example per each class. For the sake of simplicity, we call the dataset Tiny-CIFAR-10. Here, we show the complete results for the ablation experiments on all of Tiny-CIFAR-10 examples. Figure 13 shows that taking a few optimization steps is enough for the resulting images to look natural-looking. Figure 14 and 15, respectively show the effect of $\lambda_s$ and $\lambda_{TV}$ on the imperceptability of the perturbations.

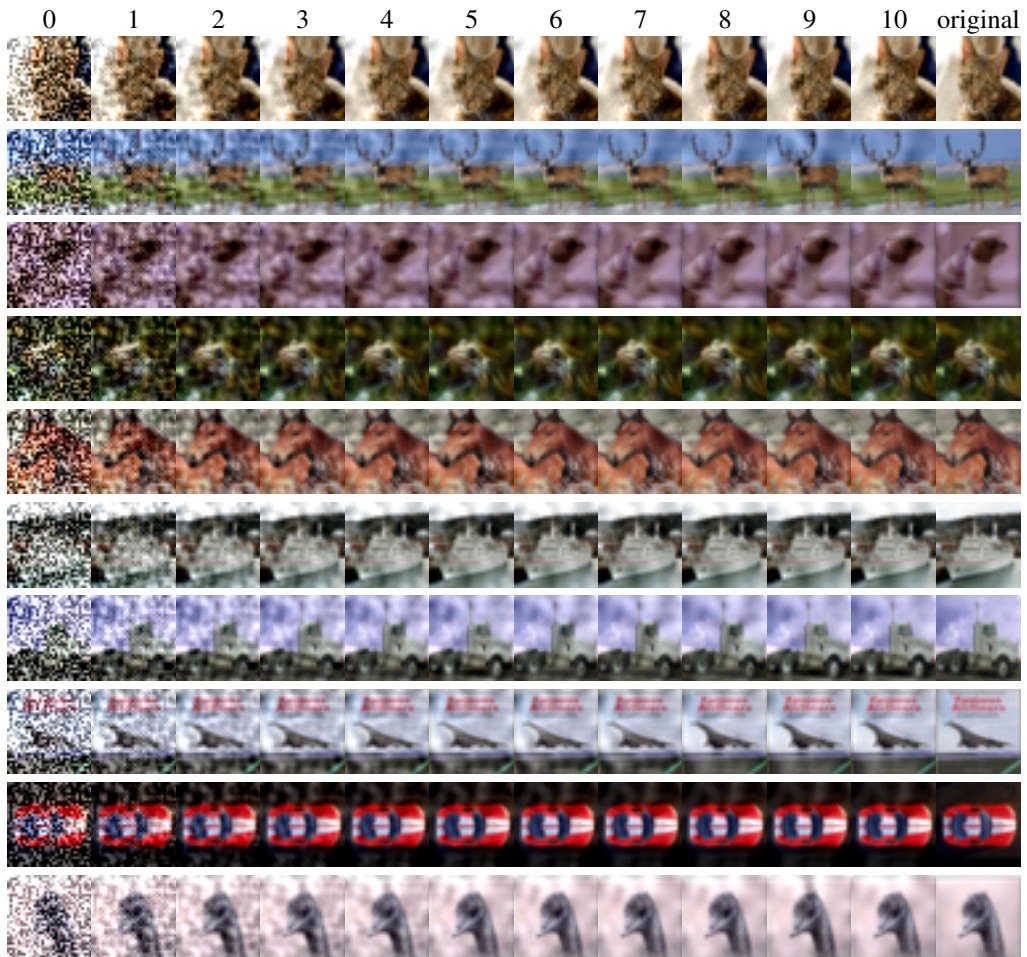

Figure 13: The first 10 steps of the optimization vs the original image for Tiny-CIFAR-10. See section 4 for the details of the experiments.

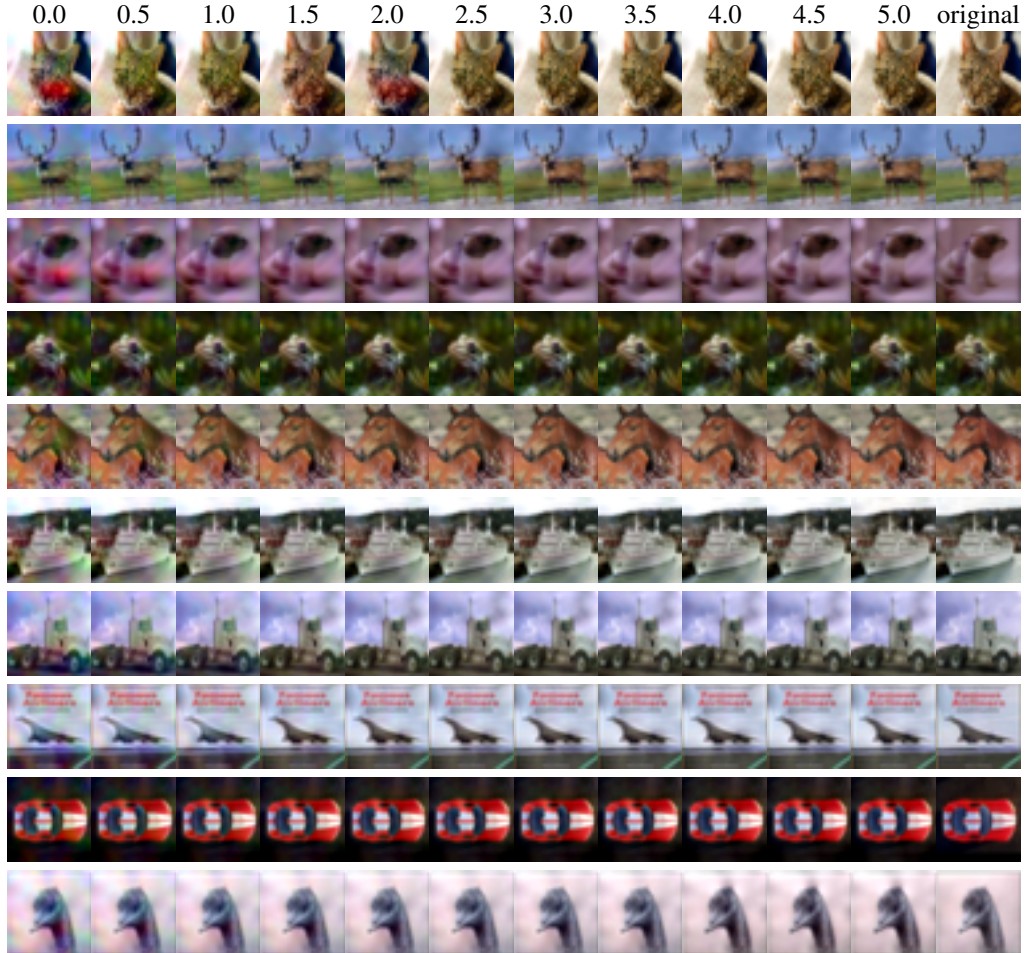

Figure 14: The visual effect of $\lambda_s$ on $Dissim(\delta)$ on Tiny-CIFAR-10. See section 4 for the details of the experiments.

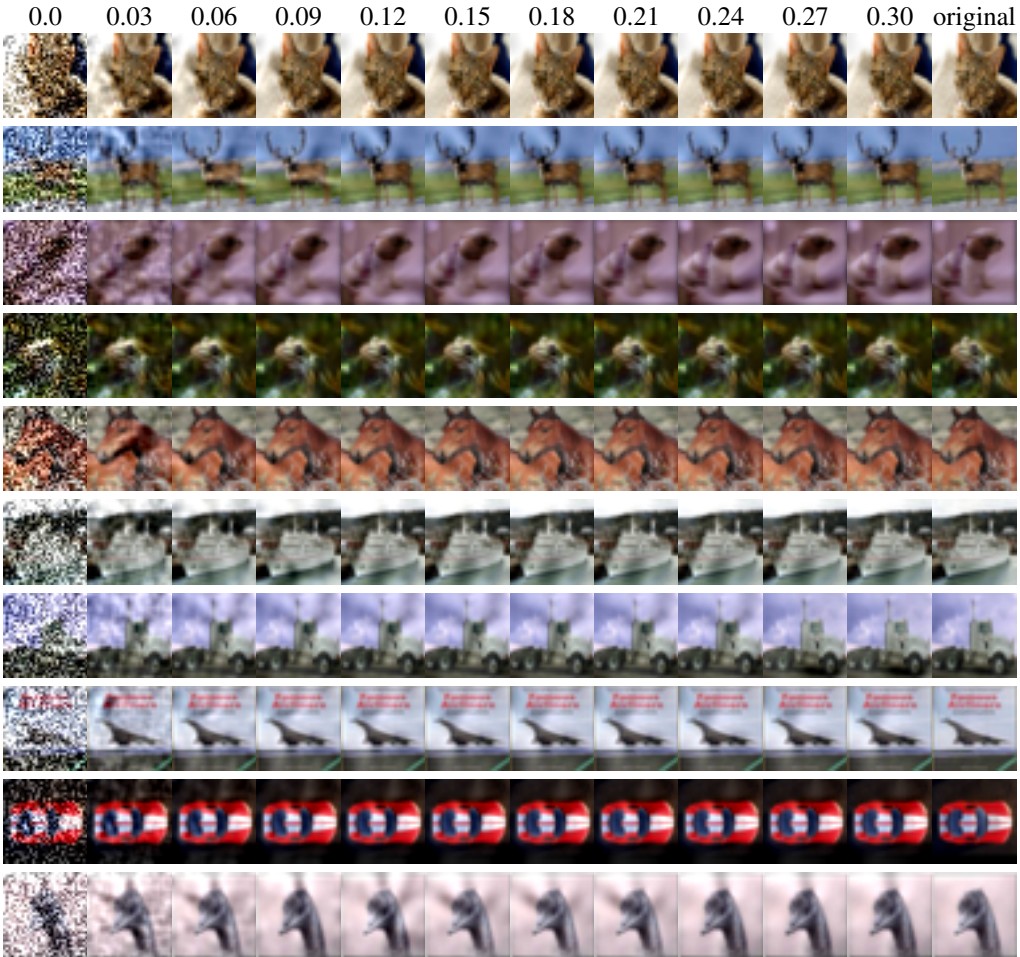

Figure 15: The visual effect of $\lambda_{tv}$ on the perturbation Tiny-CIFAR-10. See section 4 for the details of the experiments.

Table 3: Certified radii statistics produced by the Adversarially Trained Randomized Smoothing method for our adversarial examples crafted using Shadow Attack and the natural examples (larger radii are better).

| Dataset | $\sigma(l_2)$ | Adversarially Trained Randomized Smoothed | | Shadow Attack | |
| --- | --- | --- | --- | --- | --- |
| | | Mean | STD | Mean | STD |
| CIFAR-10 | 0.5 | 0.60 | 0.34 | **0.65** | 0.16 |

IMAGENET RESULTS

Many of the recent studies have explored the semantic attacks. Semantic attacks are powerful for attacking defenses (Engstrom et al., 2017; Hosseini & Poovendran, 2018; Laidlaw & Feizi, 2019). Many of semantic attacks are applicable to Imagenet, however, none of them consider increasing the radii of the certificates generated by the certifiable defenses.

Some other works focus on using generative models to generate adversarial examples (Song et al., 2018), but unfortunately none of the GAN's are expressive enough to capture the manifold of the ImageNet.

Figure 16 illustrates some of our successful examples generated by Shadow Attack to attack Randomized Smoothed classifiers for ImageNet.

## B   CERTIFICATE SPOOFING ATTACKS ON ADVERSARIALLY TRAINED SMOOTH CLASSIFIERS

Recently, Salman et al. (2019) significantly improved the robustness of Gaussian smoothed classifiers by generating adversarial examples for the smoothed classifier and training on them. In this section, we use our Shadow Attack to generate adversarial examples using the loss in eq. 4 for the SmoothAdv classifier [4]. Due to computation limitations, we attack a sample of 40% of the same validation images used for evaluating the randomized smooth classifier in section 3. The results are summarized in table 3. Comparing the results of table 1 to table 3, we can see that the SmoothAdv classifier, does produce stronger certified radii for natural examples (many of the examples in fact have the maximum radii) compared to the original randomized smoothing classifier. This can be associate to the excessive invariance introduced as a result of adversarial training. However, table 3 empirically verifies that its certificates are still subject to attacks and the certificate should not be used as a measure for robustness.

---

[4]We use the official models released at https://github.com/Hadisalman/smoothing-adversarial for attacking the SmoothAdv classifier.

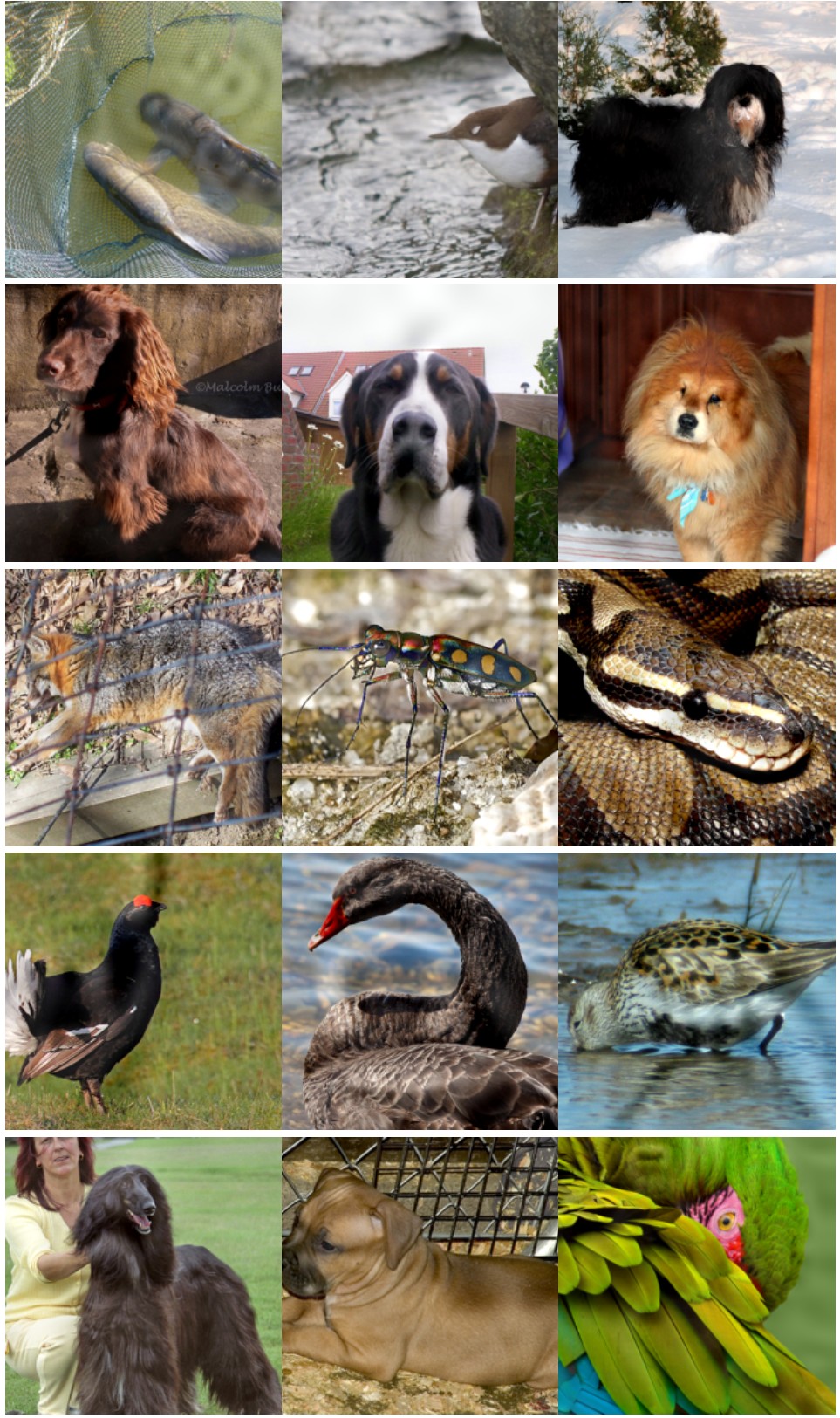

Figure 16: Natural looking Imperceptible ImageNet adversarial images which produce large certified radii for the ImageNet Gaussian smoothed classifier.

