# OpenReview forum: "BREAKING  CERTIFIED  DEFENSES:  SEMANTIC  ADVERSARIAL  EXAMPLES  WITH  SPOOFED  ROBUSTNESS  CERTIFICATES"
_ICLR.cc/2020/Conference — Accept (Poster)_

### Official Review · AnonReviewer1 · 2019-10-24
**Official Blind Review #1**

**Rating:** 6

**Review:**

The paper proposes a new way to generate adversarial images that are perturbed based on natural images called Shadow Attach. The generated adversarial images are imperceptible and have a large norm to escape the certification regions. The proposed method incorporates the quantities of total variation of the perturbation, change in the mean of each color channel, and dissimilarity between channels, into the loss function, to make sure the generate adversarial images are smooth and natural. Quantitative studies on CIFAR-10 and ImageNet shows that the new attack method can generate adversarial images that have larger certified radii than natural images. To further improve the paper, it would be great if the authors can address the following questions:

- In Table 1, for ImageNet, Shadow Attach does not always generate adversarial examples that have on average larger certified radii than the natural parallel, at least for sigma=0.5 and 1.0. Could the authors explain the reason?

- In Table 2, it is not clear to me what is the point for comparing errors of the natural images (which measures the misclassification rate of a natural image) and that of the adversarial images (which measures successful attacks rate), and why this comparison helps to support the claim that the attack results in a stronger certificates. In my opinion, to support the above claim, shouldn’t the authors provide a similar table as Table 1, directly comparing the certified radii of the natural images and adversarial images?

- From Figure 9, we see the certificate radii of the natural have at least two peaks. Though on average the certificate radii of the adversarial attacks is higher than that of the natural images, it is smaller than the right peak. Could the authors elaborate more of the results?

- Sim(delta) should be Dissim(delta) which measures the dissimilarity between channels. A smaller dissimilarity suggests a greater similarity between channels.

- Lambda sim and lambda s are used interchangeably. Please make it consistent.

- The caption of Table 1 is a little vague. Please clearly state the meaning of the numbers in the table.


**Experience Assessment:**

I do not know much about this area.

**Review Assessment: Checking Correctness Of Derivations And Theory:**

I carefully checked the derivations and theory.

**Review Assessment: Checking Correctness Of Experiments:**

I carefully checked the experiments.

**Review Assessment: Thoroughness In Paper Reading:**

I read the paper thoroughly.

---

> ### Author Response · Authors · 2019-11-15
> **Thanks for your feedback.**
>
> Thanks for your constructive feedback. We have modified the paper to clarify some of the terms per your suggestion. Please find our detailed response below:
>
> [R1: In Table 1, for ImageNet, Shadow Attack does not always generate adversarial examples that have on average larger certified radii than the natural parallel, at least for sigma=0.5 and 1.0. Could the authors explain the reason?]
>
> During attack/crafting, we need to make an adversarial example that gets misclassified even after perturbations drawn from a Gaussian distribution centered at zero with scale sigma. During evaluation, while the augmentations are drawn from a similar distribution, the realized random variables are not identical to those used for crafting the adversarial perturbation. In ImageNet, where the dimensionality is high (224X224X3) and for larger sigmas, to have a relatively dense and representative sampling, we need to sample a lot more perturbations during adversarial example crafting. However, in our experiments, we could only sample up to 400 instances per example (the maximum batch-size that could fit on our machine with 4 GPUs is 400). This results in having a sparse sample when the standard deviation is higher. One can potentially improve these results by using larger batch-sizes (i.e., sampling more) or a more powerful GPU or even a TPU, however we do not have the resources for such experiments at this time.
>
> [R1: In Table 2, it is not clear to me what is the point for comparing errors of the natural images (which measures the misclassification rate of a natural image) and that of the adversarial images (which measures successful attacks rate), and why this comparison helps to support the claim that the attack results in a stronger certificates. In my opinion, to support the above claim, shouldn’t the authors provide a similar table as Table 1, directly comparing the certified radii of the natural images and adversarial images?]
>
> In the original submission, we tried to produce tables that look like the tables in papers that we compare to. The randomized smoothing paper reports certified radii and also accuracy (1-error) under various perturbation bounds. However, the CROWN-IBP paper and the improved randomized smoothing paper based on adversarial training of smoothed classifiers (SmoothAdv) only report *error rates* using a fixed distance to the decision boundary. This is done because, unlike the Randomized Smoothing method, the radii are not directly calculated in the CROWN-IBP method and cannot be accessed directly;  CROWN-IBP takes a fixed radius chosen by the user, and either produces or fails to produce a certificate for that radius.  This is in contrast to randomized smoothing, which outputs different radii for different images (a larger radius means a stronger certificate).
>
> In regards to why we compare the errors on natural images and those of our adversarial images:  Please see the (updated) last paragraph of Section 5, which explains this comparison in detail.  In short -  we are comparing the rate at which natural images certify to the rate at which adversarial images certify.  For the case of large perturbations, we find that our adversarial image produce certificates more often than natural images!  For small perturbations, our attack still produces certificates reasonably often, although not quite as frequently as natural images. This shows that certificates alone cannot be used to reliably discern between natural images, and adversarial images produced by the proposed shadow attack.
>
> [R1: From Figure 9, we see the certificate radii of the natural have at least two peaks. Though on average the certificate radii of the adversarial attacks is higher than that of the natural images, it is smaller than the right peak. Could the authors elaborate more of the results?]
>
> This happens because, for CIFAR-10, the smoothed classifier is very “confident” on a subset of the validation images which correspond to that right peak. Here, our use of “confidence” should not be confused with the confidence of a network (output of the softmax layer). For the purpose of the certified radii, the “confidence” we are interested in is related to the prediction of the network on the Gaussian perturbed images (i.e., a very high “confident” example is an example where all of the perturbed images get the same label).
>
> [R1: Sim(delta) should be Dissim(delta) which measures the dissimilarity between channels. A smaller dissimilarity suggests a greater similarity between channels.]
> Good point! We have updated this in the revised document, and we think it enhanced clarity.
>
> [R1: Lambda sim and lambda s are used interchangeably. Please make it consistent. ]
> Fixed. Thank you.
>
> [R1: The caption of Table 1 is a little vague. Please clearly state the meaning of the numbers in the table.]
> In the revision, we have described what the numbers are representing in more detail.

---

### Official Review · AnonReviewer3 · 2019-10-24
**Official Blind Review #3**

**Rating:** 8

**Review:**

The paper presents a new attack, called the shadow attack, that can maintain the imperceptibility of adversarial samples when out of the certified radius. This work not only aims to target the classifier label but also the certificate by adding large perturbations to the image. The attacks produce a 'spoofed' certificate, so though these certified systems are meant to be secure, can be attacked. Theirs seem to be the first work focusing on manipulating certificates to attack strongly certified networks. The paper presents shadow attack, that is a generalization of the PGD attack. It involves creation of adversarial examples, and addition of few constraints that forces these perturbations to be small, smooth and not many color variations. For certificate spoofing the authors explore different spoofing losses for l-2(attacks on randomized smoothing) and l-inf(attacks on crown-ibp) norm bounded attacks.

Strengths: The paper is well written and well motivated. The work is novel since most of the current work focus on the imperceptibility and misclassification aspects of the classifier, but this work addresses attacking the strongly certified networks.

Weakness: It would be good to see some comparison to the state of the art

**Experience Assessment:**

I do not know much about this area.

**Review Assessment: Checking Correctness Of Derivations And Theory:**

I did not assess the derivations or theory.

**Review Assessment: Checking Correctness Of Experiments:**

I did not assess the experiments.

**Review Assessment: Thoroughness In Paper Reading:**

I made a quick assessment of this paper.

---

> ### Author Response · Authors · 2019-11-15
> **Thanks for your feedback.**
>
> Thank you for the encouraging review.
>
> [R3: Weakness: It would be good to see some comparison to the state of the art ]
>
> With regards to your comment on attacking the current state of the art method for smoothed classifiers, we have added new results to the resubmission (Appendix B), in which we attack the adversarially trained smooth classifier [1].
>
> [1]. Salman et al., “Provably Robust Deep Learning via Adversarially Trained Smoothed Classifiers”, NeurIPS 2019

---

### Official Review · AnonReviewer4 · 2019-11-06
**Official Blind Review #4**

**Rating:** 6

**Review:**

The paper presents a new attack: Shadow Attack, which can generate imperceptible adversarial samples. This method is based on adding regularization on total variation, color change in each channel and similar perturbation in each channel. This method is easy to follow and a lot of examples of different experiments are shown.
However, I have several questions about motivation and method.

First, the proposed attack method can yield adversarial perturbations to images that are large in the \ell_p norm. Therefore, the authors claim that the method can attack certified systems. However, attack in Wasserstein distance and some other methods can also do so. They can generate adversarial examples whose \ell_p norm is large.
I think the author should have some discussions about these related methods.

Second, I notice that compared to the result in Table 1, PGD attack can yield better results [1]. I hope to see some discussions about this. Also, Table 1 is really confused. I would not understand the meaning if I am not familiar with the experiment settings.

[1] Salman, Hadi, et al. "Provably Robust Deep Learning via Adversarially Trained Smoothed Classifiers." Neuips (2019).

**Experience Assessment:**

I have read many papers in this area.

**Review Assessment: Checking Correctness Of Derivations And Theory:**

N/A

**Review Assessment: Checking Correctness Of Experiments:**

I assessed the sensibility of the experiments.

**Review Assessment: Thoroughness In Paper Reading:**

N/A

---

> ### Author Response · Authors · 2019-11-15
> **Thanks for your feedback.**
>
> Thanks for your constructive feedback. We have modified the paper to include some of the experiments you have suggested. Please find our detailed response below:
>
> [R4: First, the proposed attack method can yield adversarial perturbations to images that are large in the \ell_p norm. Therefore, the authors claim that the method can attack certified systems. However, attack in Wasserstein distance and some other methods can also do so. They can generate adversarial examples whose \ell_p norm is large.
> I think the author should have some discussions about these related methods.]
>
> Thank you for pointing us out to the missing related work which we have included in the revision. Indeed, the Wasserstein attack and the other previously mentioned non-$\ell_p$ bounded attacks are alternatives for producing quasi-imperceptible non-$\ell_p$ bounded adversarial examples. Any of these methods can alternatively be used for generating non $\ell_p$ bounded attacks. However, one major advantage of our attack method over the Wasserstein attack may be its simplicity and scalability.
>
> Per your suggestion, we ran experiments using the Wasserstein attack. The authors of [1] suggest that the Wasserstein PGD attack works best when the attacker takes PGD steps in $ell_p$-norm directions and then project the noise back onto the Wasserstein ball. We used their official implementation and adapted it to attack the Randomized Smoothed classifier. Based on the official implementation, after every 10 iterations, if the attack is not successful, we increase the radius of the wasserstein ball in which the noise is projected back onto. Consequently, the attack is always able to reach a comparable, but slightly weaker, spoofed certified radii (~ 67% that of the shadow attack) at the cost of slightly more perceptible adversarial noise in difficult cases. Note that the reason that the examples are more perceptible than those from [1] is that they are made to produce large certified radii and not only cause misclassification (i.e., the entire Gaussian augmented batch needs to get misclassified.) A comparison of the resulting images and average certified radii of those images can be found in the following anonymized link:
> https://docs.google.com/spreadsheets/d/1F0P8aOD_5aiVjW3CrR49fudz4EgrORz7v4t0ZIJEBAo/edit?usp=sharing.
>
> [1] Wong et al., “Wasserstein Adversarial Examples via Projected Sinkhorn Iterations”.
>
>
> [R4: Second, I notice that compared to the result in Table 1, PGD attack can yield better results [1]. I hope to see some discussions about this. Also, Table 1 is really confused. I would not understand the meaning if I am not familiar with the experiment settings.]
>
> Per your request, we have attacked the work of [2] and reported results of attacking the pre-trained SmoothAdv classifiers (available in [3]) in Appendix B. Similar to the non-adversarially trained smooth classifier included in the original submission, we can produce adversarial examples for the SmoothAdv classifier which on average produce larger certified radii than their natural example counterpart. Also, in the revised document, we have expanded the caption of Table 1 to make sure that it is clear what a certified is and that a larger radii is better.
> [2]. Salman et al., “Provably Robust Deep Learning via Adversarially Trained Smoothed Classifiers”, NeurIPS 2019
> [3]. https://github.com/Hadisalman/smoothing-adversarial

---

### Decision · Program_Chairs · 2019-12-19

**Decision:**

Accept (Poster)

**Comment:**

This work presents a "shadow attack" that fools certifiably robust networks by producing imperceptible adversarial examples by search outside of the certified radius. The reviewers are generally positive on the novelty and contribution of the work.